# Retrospective Analysis of Treatment Outcomes of Maxillary Sinusitis Associated with Medication-Related Osteonecrosis of the Jaw

**DOI:** 10.3390/ijerph19127430

**Published:** 2022-06-17

**Authors:** Mitsunobu Otsuru, Saki Hayashida, Kota Morishita, Maho Murata, Sakiko Soutome, Miho Sasaki, Yukinori Takagi, Misa Sumi, Masahiro Umeda

**Affiliations:** 1Department of Clinical Oral Oncology, Nagasaki University Graduate School of Biomedical Sciences, Nagasaki 852-8588, Japan; sakihaya@nagasaki-u.ac.jp (S.H.); k-morishita@nagasaki-u.ac.jp (K.M.); mamurata@nagasaki-u.ac.jp (M.M.); mumeda@nagasaki-u.ac.jp (M.U.); 2Department of Oral Health, Nagasaki University Graduate School of Biomedical Sciences, Nagasaki 852-8588, Japan; sakiko@nagasaki-u.ac.jp; 3Department of Radiology and Biomedical Informatics, Nagasaki University Graduate School of Biomedical Sciences, Nagasaki 852-8588, Japan; sasaki-m@nagasaki-u.ac.jp (M.S.); yuki@nagasaki-u.ac.jp (Y.T.); misa@nagasaki-u.ac.jp (M.S.)

**Keywords:** antiresorptive agent, osteonecrosis, maxillary sinusitis, irrigation, treatment outcome

## Abstract

Although maxillary sinusitis often occurs in patients with medication-related osteonecrosis of the jaw (MRONJ) of the upper jaw, there have been few reports on the treatment and outcomes for maxillary sinusitis associated with maxillary MRONJ. This study aimed to retrospectively investigate the treatment outcomes of maxillary sinusitis in patients with MRONJ of the upper jaw. There were 34 patients diagnosed with maxillary MRONJ and sinusitis by preoperative computed tomography who underwent surgery in our institution between January 2011 and December 2019. Age, sex, primary disease, stage of MRONJ, class and administration period of an antiresorptive agent, corticosteroid administration, preoperative leukocyte count and serum albumin level, periosteal reaction, sinusitis grade, maxillary sinus surgical procedure, and treatment outcomes of MRONJ and sinusitis were examined. There were 7 male and 27 female patients (average age, 74.7 years). Complete healing of MRONJ was obtained in 29 of 34 patients (85.3%). Maxillary sinusitis resolved or improved in 21 patients (61.8%) but did not change or worsen in 13 patients (38.2%). We found that complete resection of necrotic bone with intraoperative irrigation of the maxillary sinus may provide good treatment outcomes for maxillary sinusitis associated with MRONJ, although our findings were not statistically significant owing to the small number of patients.

## 1. Introduction

Antiresorptive agents such as bisphosphonates or denosumab have become widely used in preventing osteoporotic fractures and treating skeletal-related events in patients with bone metastases of cancer or multiple myeloma. Antiresorptive agents may induce medication-related osteonecrosis of the jaw (MRONJ), a severe, late-onset adverse event. The treatment of MRONJ is difficult, prompting the position paper, 2014, by the American Association of Oral and Maxillofacial Surgeons (AAOMS) and the position paper, 2016, by the Japanese Society of Oral and Maxillofacial Surgeons (JSOMS) to state that the treatment goals of MRONJ are to relieve symptoms and prevent progression. These position papers recommend conservative treatment, such as antibacterial mouth rinse or oral antibiotics, as first-line therapy [1,2]. The AAOMS position paper was revised in 2022 [3], and some modifications were made to the treatment strategy. It describes that both conservative and surgical treatments are acceptable for all stages of MRONJ, depending on the patient’s situation. In contrast, some studies have drawn different conclusions. Giudice et al. reported that surgical treatment of patients in the early stages of MRONJ guarantees benefits in outcomes such as mucosal integrity and lesion downstaging, improvement in quality of life, and faster reuptake of medication therapy, especially for the oncologic patient [4]. Favia et al. also stated that MRONJ occurring both in neoplastic and non-neoplastic patients benefits more from a surgical treatment approach, whenever deemed possible, as non-surgical treatments do not seem to allow complete healing of the lesions [5]. Rupel et al. reported the first systematic review on the treatment of MRONJ in 2014, in which they stated that a surgical approach was more effective overall and in every disease stage, but more randomized controlled studies were needed to confirm this statement providing higher levels of evidence [6]. Fliefel et al. also reported in a systematic review that the treatment outcome of patients undergoing major surgery was better than those receiving conservative treatment or minimally invasive surgery [7]. A previous retrospective study also reported the superiority of surgical treatment to conservative treatment, with 361 patients with MRONJ and propensity score matching analysis that recommended surgery as the first-line treatment [8].

Although several studies have shown the superiority of surgical treatment, as described above, few studies have examined the proper extent of osteotomy. In particular, MRONJ that occurs in the maxilla often accompanies maxillary sinusitis. However, there have been few studies on appropriate treatment methods for maxillary sinusitis accompanied by MRONJ. Okuyama et al. reported that complete resection of necrotic bone is needed to achieve complete healing of maxillary MRONJ and that concomitant maxillary sinusitis tends to heal or improve clinically alongside the healing of maxillary MRONJ [9]. Sawada reported good results from simultaneously performing endoscopic sinus surgery (ESS) and necrotic bone resection [10]. However, their study had short follow-up cases and did not reveal the indications for ESS. This study aimed to investigate the treatment and prognosis of maxillary sinusitis after surgical treatment of MRONJ in the maxilla. The primary endpoint was the identification of factors related to the healing of maxillary sinusitis associated with maxillary MRONJ.

## 2. Materials and Methods

### 2.1. Study Design and Patients

This retrospective observational study was conducted in a single institute. Sixty-four patients with maxillary MRONJ underwent surgery at the Department of Oral and Maxillofacial Surgery, Nagasaki University, between January 2011 and December 2019. Among them, 34 patients diagnosed with maxillary sinusitis on preoperative computed tomography (CT) examinations were enrolled in the study.

### 2.2. Ethical Approval

The study protocol conformed to the ethical guidelines of the Declaration of Helsinki and the Ethical Guidelines for Medical and Health Research involving Human Subjects by the Ministry of Health, Labor, and Welfare of Japan. This study was approved by the Institutional Review Board (IRB) (#21021509) of Nagasaki University Hospital. As this was a retrospective study, the research plan was published on the homepage of the participating hospitals according to the instructions of the IRB, highlighting the guaranteed opt-out opportunity. Informed consent was waived by the ethics review board owing to the retrospective study design.

### 2.3. Data Examined

The following data were obtained: age, sex, primary disease (osteoporosis/malignant tumor), stage of MRONJ (AAOMS position paper, 2022 [3]), class of antiresorptive agent used (bisphosphonates/denosumab), and administration period, corticosteroid administration, diabetic status, leukocyte count and serum albumin level before surgery, periosteal reaction, maxillary sinus surgical procedure, treatment outcomes of MRONJ, and the sinusitis grade by CT examinations before and three to six months after surgery.

The treatment outcomes of MRONJ were divided into four categories: (i) complete healing, with all symptoms, including bone exposure, resolved; (ii) partial healing, defined as downstaging; (iii) no change or improvement in clinical signs; and (iv) progression of the disease and upstaging during observation. Sinusitis was graded according to the method reported by Kurabayashi [11]. Grade 1: sinusitis was limited to the floor of the maxillary sinus. Grade 2: sinusitis was spread over the entire maxillary sinus from the floor to the top, but an air cavity remained in the sinus. Grade 3: sinusitis has spread to the whole maxillary sinus. Grade 4: inflammation has spread to the ethmoid sinus (Figure 1).

Multi-slice CT examinations were performed before and three to six months after surgery. The diagnosis of maxillary sinusitis using CT images was made by the consensus of three oral surgeons and two radiologists.

### 2.4. Statistical Analysis

All statistical analyses were performed using the SPSS software (version 24.0; Japan IBM Co., Ltd., Tokyo, Japan). The correlation between each variable and surgical outcome was analyzed using the Mann–Whitney U test for continuous variables and Fisher’s exact test for categorical variables. A two-tailed probability of less than 0.05 was considered significant.

## 3. Results

### 3.1. Patient Characteristics

The background characteristics of the patients are listed in Table 1. The participants of this study were 7 men and 27 women, with an average age of 74.7 years. The primary disease was osteoporosis in 17 patients and malignant tumors in 17 patients. In all, 11 patients had stage 1–2 MRONJ, while 23 patients had stage 3 MRONJ. Moreover, 7 patients had preoperative grade 1 maxillary sinusitis, 9 patients had grade 2, 1 patient had grade 3, and 17 had grade 4.

### 3.2. Treatment and Outcome of MRONJ

All patients underwent removal of the necrotic bone and the surrounding healthy bone, and the wound was closed with a mucoperiosteal flap. If necessary, the periosteum on the buccal side was incised and then primarily sutured. No cases used a buccal fat pad pedicle flap.

All symptoms resolved after surgery in 29 of 34 patients (85.3%), while 4 patients showed partial remission, and 1 patient showed no change. None of the patients had progressive diseases. The cumulative cure rates at 1-, 2-, and 3-year follow ups were 81.1%, 85.9%, and 95.3%, respectively (Figure 2).

### 3.3. Procedures for Maxillary Sinus and Outcome of Sinusitis

For the procedure in the maxillary sinus of 21 patients, the wound was closed after the removal of the necrotic bone without treatment for the sinusitis. In 13 patients, the sinus mucosa was excised in a small area, after the necrotic bone was removed, and then the maxillary sinus was irrigated with normal saline before closing the wound (Figure 3) (Table 2). Irrigation is often performed in patients with advanced-stage maxillary sinusitis. There were no cases of residual oro-antral fistulas.

Antibacterial drug administration was performed immediately before the operation and for about 2 days after the operation as in the case of other oral surgery, but it was often performed for a long period depending on the infection situation. In some cases with severe maxillary sinusitis, amoxicillin and clarithromycin were administered for 1 to 3 months after surgery.

Maxillary sinusitis healed in 8 patients, improved in 13, remained unchanged in 10, and worsened in 3 (Figure 4). The cases of severe maxillary sinusitis (grades 3–4) decreased from 18 before surgery to 6 after surgery (Figure 4 and Figure 5). Figure 4 shows the treatment results based on the preoperative grade of the maxillary sinusitis. In three patients who had grade 1 sinusitis before surgery, maxillary sinusitis worsened after surgery. Furthermore, 7 of 9 grade 2 patients and 13 of 18 grade 3–4 patients healed or improved. Five patients showed no change postoperatively.

### 3.4. Factors Related to Treatment Outcome of Maxillary Sinusitis

We examined the differences in the outcomes of maxillary sinusitis between 21 patients who healed or improved and 13 patients who remained unchanged or worsened. Maxillary sinusitis was cured or improved in 20 of the 29 patients (69.0%) in whom MRONJ was cured, while it was cured or improved in only 1 of the 5 patients (20%) in whom MRONJ was not cured. In addition, maxillary sinusitis was resolved or improved in 10 of 13 patients (76.9%) who received sinus irrigation and in 11 of 21 patients (52.4%) who did not receive irrigation (Figure 6). However, there was no significant association between each variable and maxillary sinusitis outcome, possibly because of the small population size (Table 3).

## 4. Discussion

This study showed that about half of the patients with maxillary MRONJ concomitantly had maxillary sinusitis, even in stage 1–2 patients for whom the necrotic bone did not extend to the floor of the maxillary sinus, and that intraoperative irrigation therapy at the same time as necrotic bone resection was effective against maxillary sinusitis.

Maxillary sinusitis may cause infection not only in the maxillary sinus but also in the other paranasal sinuses and eventually systemic infection. Therefore, we believe that the treatment goal for maxillary MRONJ, beyond the control of MRONJ, is the control of maxillary sinusitis as well.

Regarding the treatment method of MRONJ, non-surgical treatment was recommended previously. Since the invasive procedure on the bone can result in new osteonecrosis, it was recommended not to expose the surrounding healthy bone during necrotic resection [12]. On the other hand, there have been many reports that surgical treatment has better outcomes. As a result of our search, we found 18 case series papers covering more than 100 cases [4,5,8,13,14,15,16,17,18,19,20,21,22,23,24,25,26,27]. Among them, only 4 papers recommended conservative therapy as a treatment method for MRONJ [13,14,15,16], and 13 papers recommended surgical treatment [4,5,8,17,18,19,20,21,22,23,24,25,26,27]. Furthermore, five papers [8,18,19,23,26] performing multivariate analysis reported significantly better outcomes for patients treated surgically than for those treated by conservative therapy. As described above, it has become common to recommend surgery as a treatment method for MRONJ. However, the majority of these reports were related to patients with MRONJ in the mandible, and few papers described in detail how to treat maxillary cases [9,10].

As for the concomitant rate of maxillary sinusitis, Mast et al. [28] reported that 23 of 53 (43.6%) patients with MRONJ of the maxilla showed signs of maxillary sinusitis, and Sawada S, et al. [7] reported that 31 of 68 (45.6%) patients with maxillary MRONJ showed maxillary sinusitis by preoperative CT examinations. This indicates that one-third of the patients developed maxillary sinusitis even though the necrotic bone did not extend to the maxillary sinus.

There are various reports on the treatment of maxillary sinusitis associated with maxillary MRONJ. Matsumoto et al. [29] reported 16 patients with maxillary sinusitis associated with MRONJ. Sequestrectomy was performed in all cases, and conservative treatment was performed for maxillary sinusitis. Opacification in the sinuses improved after treatment in 10 patients, partially improved in 3 patients, and remained unchanged in 2 patients, and imaging assessment after treatment could not be conducted for 1 patient. Maurer et al. [30] reported 10 MRONJ patients with concomitant maxillary sinusitis. Recurrence of MRONJ was seen in four cases, and maxillary sinusitis also recurred in all of them. Maxillary sinusitis was also cured in all six cases in which MRONJ was cured. In the case of antrotomy, maxillary sinusitis also recurred when MRONJ relapsed. Park et al. [31] reported that 38 of 62 patients (61.3%) with maxillary MRONJ showed maxillary sinusitis, and among them, 27 patients (43.5%) had advanced MRONJ with bony destruction of the sinus floor. After sequestrectomy, leukocyte-rich and platelet-rich fibrin (L-PRF) and recombinant human bone morphogenic protein-2 (rhBMP-2) were inserted selectively according to the consent of the patients. For sinus management of 38 patients with maxillary sinusitis, they performed sequestrectomy only in 16 patients and a combination surgery of sequestrectomy and functional endoscopic surgery (FESS) in 22 patients. In MRONJ with maxillary sinusitis patients, there were no differences in treatment outcome according to FESS performance. A higher percentage of MRONJ resolution was seen in the patients treated by both L-PRF and rhBMP-2 insertion as compared to sequestrectomy only or L-PRF insertion, but it was not statistically significant. When limited to advanced cases, at 4 months postoperatively, the percentage of MRONJ resolution was 84.2% in the patients treated by FESS, which was statistically significant when compared to the MRONJ resolution in the patients who did not receive FESS (37.5%). Cano-Durán et al. also reported the effect of L-PRF in the surgery for maxillary MRONJ [32]. Furthermore, there are reports on the usefulness of some surgical methods for MRONJ that has progressed to the maxillary sinus. Berrone et al. [33] and Jose et al. [34] reported sequestrectomy and reconstruction using a pedicled buccal fat pad for stage 3 MRONJ of the maxilla. As mentioned above, conservative therapy, ESS, antrotomy, use of L-PRF or BMP, buccal fat pad flap, etc., have been reported as treatments for maxillary sinusitis associated with MRONJ, but it is not clear which treatment is superior.

Sawada reported that four patients with maxillary MRONJ with sinusitis achieved healing by resection of necrotic bone combined with ESS [10]. Park reported that ESS showed a higher percentage of resolution of MRONJ, as described above [31]. However, the combined use of ESS may increase surgical invasiveness. Furthermore, their study was observational rather than interventional, and it was difficult to judge the effectiveness of ESS for maxillary sinusitis associated with MRONJ. It is widely accepted that wounds need to be closed during MRONJ surgery. In our previous study of tooth extraction in patients treated with antiresorptive agents, open wounds became one of the risk factors for developing MRONJ [35]. Resection of the maxillary sinus mucosa by ESS opens the maxillary bone surface into the maxillary sinus, which may cause local infection by maxillary sinus bacteria and worsen MRONJ lesions. In addition, postoperative administration of antibiotics does not improve the prognosis of maxillary sinusitis because of inadequate antibiotic coverage from poor blood flow in MRONJ-induced osteosclerosis. Based on these facts, we have not yet determined whether it is appropriate to perform ESS for maxillary sinusitis associated with maxillary MRONJ.

The basis of MRONJ surgery is to remove the infected bone. Surgical results have been reported to be better with extensive surgery, which removes necrotic bone and surrounding healthy bone as a safety margin, than with conservative surgery, which removes only necrotic bone. However, on the other hand, there are many reports that good treatment results were obtained by adding adjuvant therapy to conservative surgery. As an adjuvant therapy, hyperbaric oxygen therapy (HBO) [36], low-level laser therapy (LLLT) [37], platelet-rich plasma (PRP) [38], platelet-rich fibrin (PRF) [39], bone morphogenetic protein (BMP) [40], ozone [41], and teriparatide [42], mesenchymal stem cells (MSCs) [43], and human umbilical cord-derived mesenchymal stem cells (hUC-MSCs) [44] have been applied in combination with surgery, and the results seem to be effective. As mentioned above, there is a report that L-PRF and rhBMP-2 was used as an adjunct therapy for maxillary MRONJ, and better surgical results were obtained than with surgery alone [31]. However, many of these reports were case–control studies, with a few dealing with cases without controls, and the evidence remains questionable.

For odontogenic maxillary sinusitis, maxillary sinus irrigation therapy has been performed along with antibacterial drug therapy for a long time [45]. There are two methods, one is to pierce the maxillary sinus from the nasal cavity and clean it, and the other is to perform mucosal incision and bone excision from the canine fossa in the oral cavity. The latter is easy to operate and is often used in the field of oral surgery [46,47]. Applying this method, during the maxillary MRONJ operation, after resection of the necrotic bone, a part of the maxillary sinus mucosa is incised from the oral cavity, the maxillary sinus is punctured, and the maxillary sinus is thoroughly washed with physiological saline, followed by primary suture with a mucoperiosteal flap. Although the number of cases was small and it was not possible to compare it with the treatment results of ESS, it was suggested that intraoperative maxillary sinus irrigation therapy is effective as one of the surgical methods for maxillary sinusitis associated with MRONJ. However, since there were cases in which maxillary sinusitis could not be cured with just one intraoperative irrigation, so it was considered necessary to continue some kind of treatment for maxillary sinusitis in the future.

Some issues need to be clarified regarding the maxillary MRONJ. We reported that MRONJ often showed periosteal reactions on CT images and that periosteal reactions were significantly associated with postoperative recurrence [48,49,50]. However, in the maxillary cases examined in this study, the presence of periosteal reaction did not affect the prognosis of MRONJ and was not associated with the treatment outcome of maxillary sinusitis. The periosteal reaction observed in maxillary MRONJ may have different biological activity than mandibular MRONJ.

This study has some limitations. First, because this was a single-center, small-scale study, it is unclear whether the results can be generalized. Second, there was difficulty in establishing a stable, long-term follow up of the patients with MRONJ owing to primary and underlying diseases. However, this is the first study to examine the treatment and outcomes of maxillary sinusitis often seen in MRONJ. In the future, we would like to further increase the number of cases and establish an appropriate treatment method for maxillary sinusitis.

## 5. Conclusions

This study demonstrated that complete resection of necrotic bone and intraoperative irrigation of the maxillary sinus may provide good treatment outcomes for maxillary sinusitis associated with MRONJ, although not the results were not statistically significant owing to the small number of patients. In future research, a multicenter study using different treatment methods should be conducted for better generalizability of information.

## Figures and Tables

**Figure 1 ijerph-19-07430-f001:**
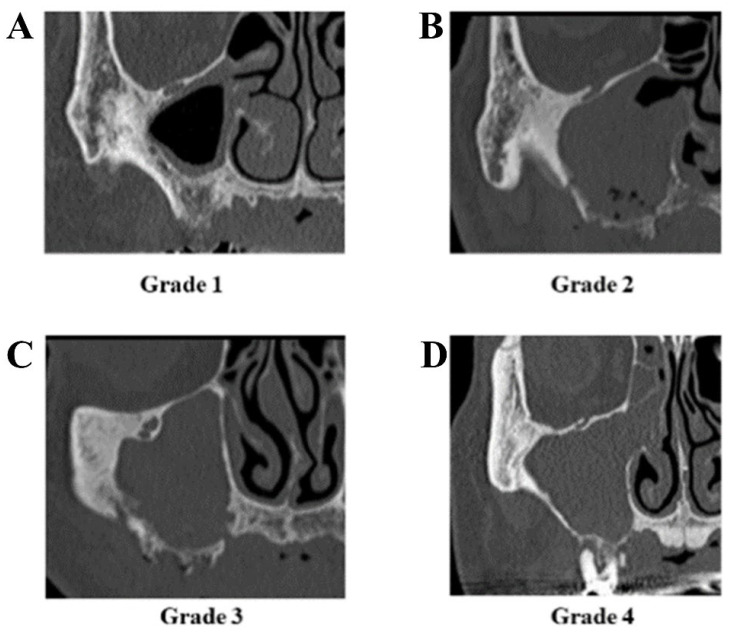
Grade of maxillary sinusitis. (**A**) Limited to the floor of the sinus (grade 1), (**B**) spreading over the entire maxillary sinus, but an air cavity remains (grade 2), (**C**) spreading to the whole maxillary sinus (grade 3), (**D**) spreading to ethmoid sinus (grade 4).

**Figure 2 ijerph-19-07430-f002:**
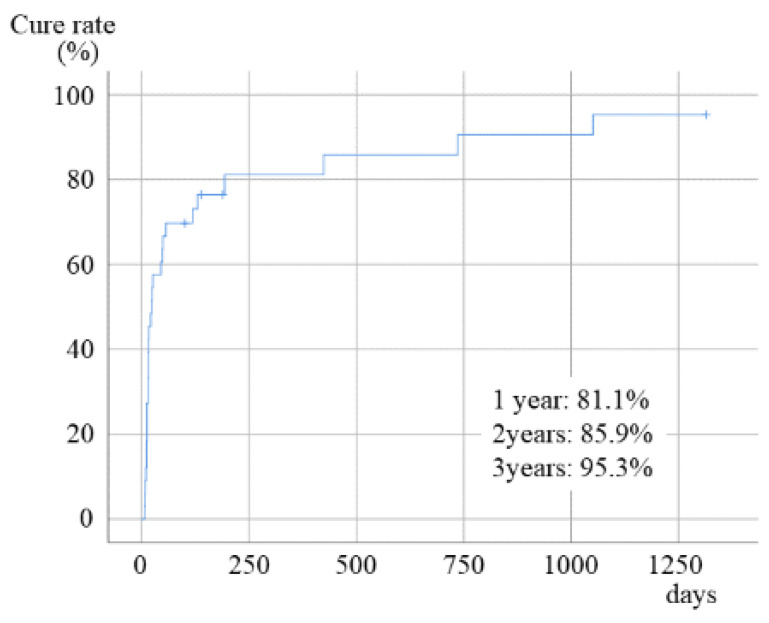
Healing rate of MRONJ after surgery.

**Figure 3 ijerph-19-07430-f003:**
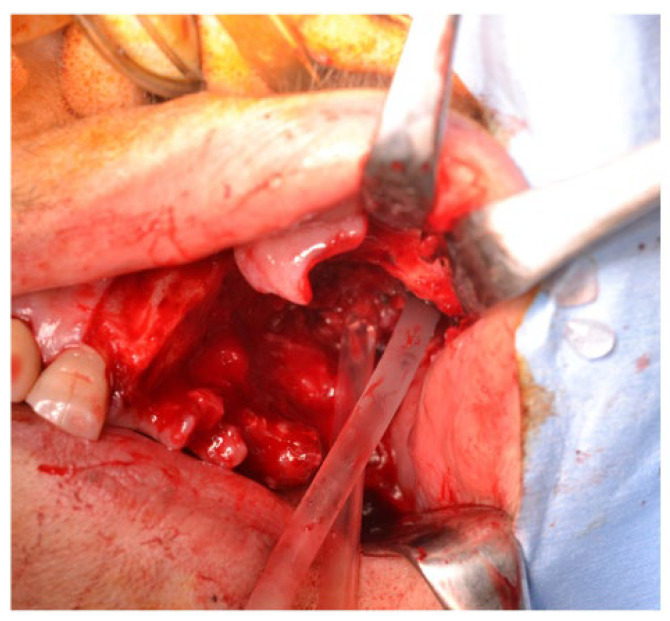
Irrigation of maxillary sinus during surgery.

**Figure 4 ijerph-19-07430-f004:**
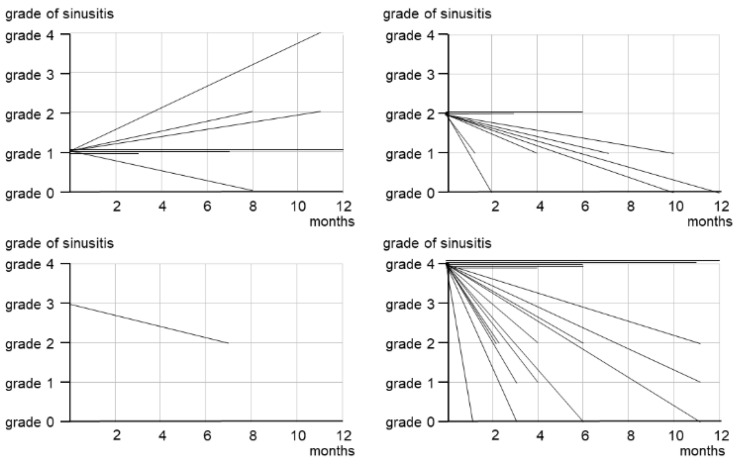
Treatment outcomes of maxillary sinusitis by initial grade.

**Figure 5 ijerph-19-07430-f005:**
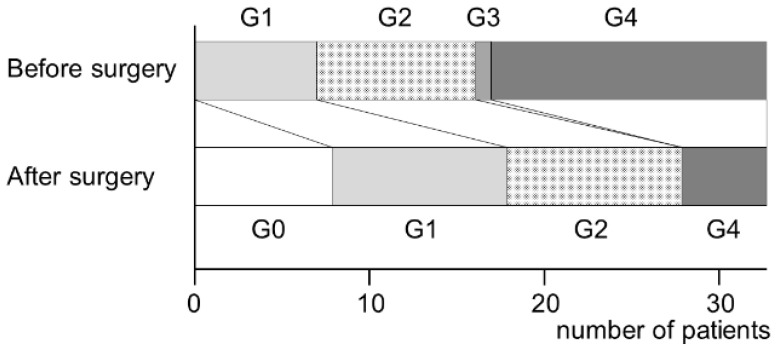
Grade changes of maxillary sinusitis before and after surgery.

**Figure 6 ijerph-19-07430-f006:**
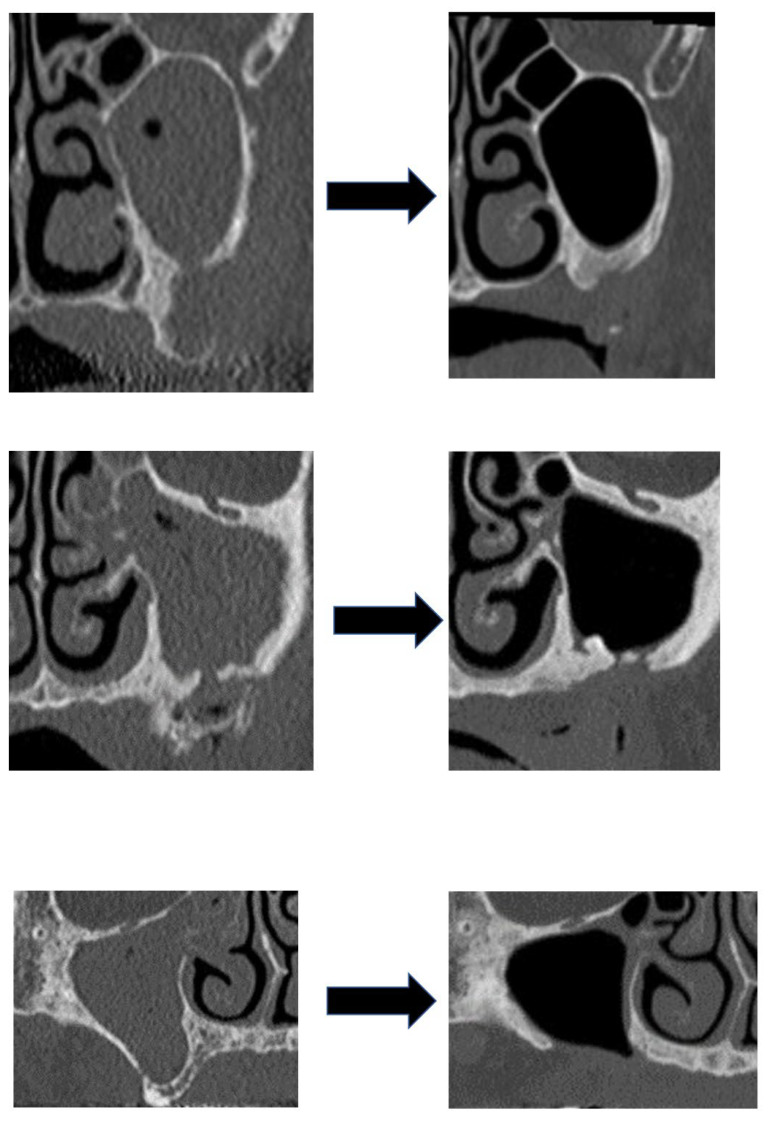
Three cases of maxillary sinusitis healing by intraoperative irrigation.

**Table 1 ijerph-19-07430-t001:** Patient characteristics.

Variable		Number of Patients/Mean ± Standard Division
Sex	male	7
	female	27
Age	(years)	74.7 ± 11.7
Primary disease	osteoporosis	17
	malignant tumor	17
Stage of MRONJ	stage 1	1
	stage 2	10
	stage 3	23
Antiresorptive agent	BP	23
	Dmab	7
	BP → Dmab	4
Duration of administration	(weeks)	59.8 ± 37.0
Use of corticosteroid	(−)	26
	(+)	8
Diabetes	(−)	28
	(+)	6
Leukocyte	(×10^3^/μL)	6.77 ± 2.86
Albumin	(g/dL)	3.50 ± 0.45
Sequester separation	(−)	19
	(+)	15
Periosteal reaction	(−)	25
	(+)	9
Postoperative antibiotics	(−)	29
	(+)	5
Grade of sinusitis	grade 1	7
	grade 2	9
	grade 3	1
	grade 4	17
Total		34

**Table 2 ijerph-19-07430-t002:** Procedures for maxillary sinus.

Procedures for Maxillary Sinus	Grade of Maxillary Sinusitis				Total
	Grade 1	Grade 2	Grade 3	Grade 4	
None	7	6	1	7	21
Intraoperative irrigation	0	3	0	10	13
Total	7	9	1	17	34

**Table 3 ijerph-19-07430-t003:** Factors related to treatment outcome of maxillary sinusitis (univariate analysis).

Variable		The Outcome of Maxillary Sinusitis		*p*-Value
		No Change/Worsening	Improvement	
Sex	male	2	5	0.555
	female	11	16	
Age	(years)	69.7 ± 12.2	77.7 ± 10.5	0.299
Primary disease	osteoporosis	5	12	0.241
	malignant tumor	8	9	
Stage of MRONJ	stage 1–2	7	4	0.042
	stage 3	6	17	
Antiresorptive agent	BP	7	16	0.164
	Dmab/BP → Dmab	6	5	
Duration of administration	(weeks)	52.9 ± 32.0	64.5 ± 40.2	0.090
Use of corticosteroid	(−)	9	17	0.352
	(+)	4	4	
Diabetes	(−)	11	17	0.584
	(+)	2	4	
Leukocyte	(×10^3^/μL)	6.68 ± 2.38	6.83 ± 3.19	0.825
Albumin	(g/dL)	3.40 ± 0.41	3.56 ± 0.47	0.679
Sequester separation	(−)	9	10	0.191
	(+)	4	11	
Periosteal reaction	(−)	10	15	0.525
	(+)	3	6	
Postoperative antibiotics	(−)	11	18	0.647
	(+)	2	3	
Grade of sinusitis	grade 1–2	8	8	0.164
	grade 3–4	5	13	
Procedure to the maxillary sinus	none	10	11	0.143
	irrigation	3	10	
Outcome of MRONJ	non-healing	4	1	0.059
	healing	9	20	
Total		13	21	

## Data Availability

Not applicable.

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
