# Peer review of "Retrospective Analysis of Treatment Outcomes of Maxillary Sinusitis Associated with Medication-Related Osteonecrosis of the Jaw"

_ijerph, 2022, doi:10.3390/ijerph19127430_

Round 1
Reviewer 1 Report
- The English is entirely correct and satisfactory.
- Summary : This paper is a retrospective study looking at treatment and outcomes of 34 patients with maxillary sinusitis when the maxilla has been/is affected by Medication-Related Osteonecrosis ( MRONJ ), From 2011 through 2019. Complete healing of MRONJ occurred in 85.3% of MRONJ-affected maxillae after surgery. Sinusitis resolved or improved in 61.8% and there was no worsening in the remainder. The authors advocate resection of involved maxillary bone.
- Strengths : Supports surgery as superior to conservative management. Sinus irrigation gave better results than no irrigation.
- Weaknesses : Small sample size. Single center. Difficulty of doing long-term follow up.
- Major revisions : None.
- Minor revisions : Could do with a bit more explanation of the surgical technique e.g buccal mucosal flap for both bone removal and access to the maxillary sinus. Please say if any patients had persisting oro-antral communications ( oro-antral fistulas ).
Overall I think this is a well-written and useful paper and I support it being published.
Author Response
To Reviewer #1
Could do with a bit more explanation of the surgical technique e.g buccal mucosal flap for both bone removal and access to the maxillary sinus. Please say if any patients had persisting oro-antral communications ( oro-antral fistulas ).
(Reply)
Line 133-136: The sentences “All patients underwent removal of necrotic bone and the surrounding healthy bone, and the wound was closed with a mucoperiosteal flap. If necessary, the periosteum on the buccal side was incised and then primarily sutured. No cases were using buccal fat pad pedicle flap.” was added.
Line 149: “There were no cases of residual oro-antral fistula.” was added.
Reviewer 2 Report
Manuscript ID: ijerph- 1736487
Title: Retrospective Analysis of Treatment Outcomes of Maxillary Sinusitis Associated with Medication-Related Osteonecrosis of the Jaw
The manuscript entitled “Retrospective Analysis of Treatment Outcomes of Maxillary Sinusitis Associated with Medication-Related Osteonecrosis of the Jaw” submitted to IJERPH aims to retrospectively investigate the treatment outcomes of maxillary sinusitis in patients with MRONJ of the upper jaw.
Although the manuscript is on a not very large case series, this retrospective study may suggest further clinical case approaches of patients with MRONJ.
I have some suggestions to improve deeply the quality of the manuscript, enriching the text with further notions.
English form: I suggest to perform a check of English text form
Abstract: Please, structure in a better way this section.
Introduction:
- Reference n.1; I strongly suggest referring to the latest AAOMS guidelines for the diagnosis and treatment of ONJ pathology [PMID: 35300956]
- When many studies on the advantages and disadvantages of the surgical approach are listed in the introduction, I note that the advantages of performing a minimally invasive surgical approach in the early stages of this disease have not been mentioned. As suggested in the new position paper of the AAOMS I suggest to include these documented notions in the following scientific articles [PMID: 32615096 - PMID: 29480596].
- I suggest to include a brief part about new possible correlation of BPs and exosomes or mesenchymal stem cells in the onset of MRONJ disease [PMID: 33255626 - PMID: 34347277]
- I suggest to specify the primary outcome variable at the end of introduction part.
Methods:
- When referring to the AAOMS classification for the disease station, I suggest using the latest guidelines (2022).
- Figure 1: the CT cuts in Figure 1 are not clear. I strongly suggest that you include images of different patients, as enrolled in your retrospective study, to provide more clarity for readers and the scientific community.
- What type of CT was performed on each patient?
It seems complex to me that the ethics committee agreed to perform 3 multislice CTs in 6 months....
Results:
- “This indicates that 112 one-third of the patients developed maxillary sinusitis even though the necrotic bone did 113 not extend to the maxillary sinus.”
I suggest to include this sentence in the discussion part; please, delete it from results part.
- When patients had sinusitis involving the ethmoid sinuses, was ENT consultation required? In all cases, was there resolution after oral surgery?
- Was irrigation of the sinus site performed with saline only?
- Why were no antibiotic washings used?
- What medical treatment did the patients receive?
I strongly suggest including these notions in the methods section.
The management of sinus involvement is multidisciplinary and may involve various surgical and pharmacological therapeutic approaches.
- Figure 6: As mentioned above, I suggest including more cuts of the CT to enable the reader and the scientific community to appreciate the results obtained in this study.
It is important to add more images of different patients to appreciate the goodness of the therapy performed.
Discussion:
- For the treatment of onj involving the maxillary sinus, it is interesting to highlight how the use of PRF membranes or the bichat fat bubble can find efficacy in the management of complex onj cases [PMID: 28936298 - PMID: 33433526 - PMID: 26183009 - PMID: 34028628]. I suggest adding part of the discussion by correlating it with the present literature.
Conclusion:
- The conclusions do not seem to me to be very much in line with the results obtained; to speak only of irrigation of the surgical site seems to me to be very reductive, not giving importance to the surgical therapy and the margins to be considered in order to increase the success or downstaiging rates of the pathology.
I suggest the authors revise this part thoroughly.
Although the manuscript may offer insights into the surgical management of onj cases involving the maxillary sinus, I suggest a second review as the authors need to clarify some aspects of the methods and iconography.
Furthermore, I believe the references section needs to be implemented by adding recent studies on this pathology.
Author Response
To Reviewer #2
- English form: I suggest to perform a check of English text form.
(Reply)
A check of the English test form was performed.
- Reference n.1; I strongly suggest referring to the latest AAOMS guidelines for the diagnosis and treatment of ONJ pathology [PMID: 35300956]
When many studies on the advantages and disadvantages of the surgical approach are listed in the introduction, I note that the advantages of performing a minimally invasive surgical approach in the early stages of this disease have not been mentioned. As suggested in the new position paper of the AAOMS I suggest to include these documented notions in the following scientific articles [PMID: 32615096 - PMID: 29480596].
I suggest to include a brief part about new possible correlation of BPs and exosomes or mesenchymal stem cells in the onset of MRONJ disease [PMID: 33255626 - PMID: 34347277]
(Reply)
We quoted the papers you showed us (reference #3, 4, 5, 43, 44), and descriptions of these papers were added in Introduction and Discussion.
Line 43-53: “The AAOMS Position Paper has been revised in 2022 [3], and some modifications were made to the treatment strategy. It describes that both conservative and surgical treat-ments are acceptable for all stages of MRONJ, depending on the patient's situation. In contrast, some studies have drawn different conclusions. Giudice et al. reported that surgical treatment of patients in the early stages of MRONJ guarantees benefits in outcomes such as mucosal integrity and lesion downstaging, improvement in quality of life, and faster reuptake of medication therapy, especially for the oncologic patient [4]. Favia et a. also stated that MRONJ occurring both in neoplastic and non-neoplastic pa-tients benefits more from a surgical treatment approach, whenever deemed possible, as non-surgical treatments do not seem to allow complete healing of the lesions [5].”
Line 263-264: “mesenchymal stem cells (MSCs) [43], and human umbilical cord-derived mesenchymal stem cells (hUC-MSCs) [44]” was added.
- I suggest to specify the primary outcome variable at the end of introduction part.
(Reply)
Line 72-74: The sentence “The primary endpoint was the identification of factors related to the healing of maxillary sinusitis associated with maxillary MRONJ” was added at the end of Introduction.
- When referring to the AAOMS classification for the disease station, I suggest using the latest guidelines (2022).
(Reply)
Line 93: “stage of MRONJ (AAOMS Position Paper 2014 [1])” was revised to “stage of MRONJ (AAOMS Position Paper 2022 [3])”.
- Figure 1: the CT cuts in Figure 1 are not clear. I strongly suggest that you include images of different patients, as enrolled in your retrospective study, to provide more clarity for readers and the scientific community.
(Reply) Figure 1 was changed to a clearer image.
- What type of CT was performed on each patient?
(Reply)
Line 110: “CT examinations” was revised to “Multi-slice CT examinations”.
- It seems complex to me that the ethics committee agreed to perform 3 multislice CTs in 6 months....
(Reply) CT was taken before surgery, 3-6 months after surgery, and thereafter depending on the symptoms. This is not a research purpose, but a routine clinical practice. This applies not only to MRONJ but also to other disorders. For example, in the case of oral cancer, contrast-enhanced CT is taken every 3 months for 2 years after surgery. Japan has the largest number of CTs per hospital in the world, and CTs are taken relatively frequently.
- “This indicates that one-third of the patients developed maxillary sinusitis even though the necrotic bone did not extend to the maxillary sinus.”
I suggest to include this sentence in the discussion part; please, delete it from results part.
(Reply)
Line 207-209: The sentence “This indicates that one-third of the patients developed maxillary sinusitis even though the necrotic bone did not extend to the maxillary sinus” was moved to Discussion.
- When patients had sinusitis involving the ethmoid sinuses, was ENT consultation required? In all cases, was there resolution after oral surgery?
(Reply)
Figure 4 shows the grades of preoperative and postoperative maxillary sinusitis. Maxillary sinusitis healed in 8 patients, improved in 13, remained unchanged in 10, and worsened in 3 by CT examinations. Depending on the severity of clinical symptoms and CT findings, we consult with an otolaryngologist as appropriate.
- Was irrigation of the sinus site performed with saline only.
(Reply)
It may be more desirable to perform a sinus irrigation with a drug such as an antibacterial drug or chlorhexidine. However, local irrigation with antibacterial agents is not permitted in Japan, and the application of 0.12% chlorhexidine, which is often used overseas, to the mucous membrane is contraindicated.
- Why were no antibiotic washings used?
(Reply)
As mentioned above, irrigation with antibacterial drugs is not permitted by public health insurance system in Japan. We would like to consider using antibacterial drugs outside the insurance coverage in the future.
- What medical treatment did the patients receive? I strongly suggest including these notions in the methods section. The management of sinus involvement is multidisciplinary and may involve various surgical and pharmacological therapeutic approaches.
(Reply)
Line 150-155: The following sentences were added.
Antibacterial drug administration was performed immediately before the opera-tion and for about 2 days after the operation as in the case of other oral surgery, but it was often performed for a long period depending on the infection situation. In some cases with severe maxillary sinusitis, amoxicillin and clarithromycin were administered for 1 to 3 months after surgery.
- Figure 6: As mentioned above, I suggest including more cuts of the CT to enable the reader and the scientific community to appreciate the results obtained in this study. It is important to add more images of different patients to appreciate the goodness of the therapy performed.
(Reply)
Figure 6 was revised to include more images.
- Discussion: For the treatment of onj involving the maxillary sinus, it is interesting to highlight how the use of PRF membranes or the bichat fat bubble can find efficacy in the management of complex onj cases [PMID: 28936298 - PMID: 33433526 - PMID: 26183009 - PMID: 34028628]. I suggest adding part of the discussion by correlating it with the present literature.
(Reply)
Of the 4 references presented by the Reviewer, 3 (PMID: 28936298, 26183009, and 34028628) were cited (reference #33, 34, 35), excluding 1 (PMID: 33433526) which is a case report of the mandible.
Line 233-240: The sentence “As mentioned above, conservative therapy, ESS, antrotomy, etc. have been reported as treatments for maxillary sinusitis associated with MRONJ, but it has not been clear which treatment is superior.” was revised to “Cano-Durán et al. also reported the effect of L-PRF in the surgery for maxillary MRONJ [32]. Furthermore, there are reports on usefulness of some surgical methods for MRONJ that have progressed to the maxillary sinus. Berrone et al. [33] and Jose et al. [34] re-ported sequestrectomy and reconstruction using a pedicled buccal fat pad for stage 3 MRONJ of the maxilla. As mentioned above, conservative therapy, ESS, antrotomy, use of L-PRF or BMP, buccal fat pad flap, etc. have been reported as treatments for maxillary sinusitis associated with MRONJ, but it has not been clear which treatment is superior.”.
- The conclusions do not seem to me to be very much in line with the results obtained; to speak only of irrigation of the surgical site seems to me to be very reductive, not giving importance to the surgical therapy and the margins to be considered in order to increase the success or downstaiging rates of the pathology. I suggest the authors revise this part thoroughly.
(Reply)
The conclusion was revised as follows.
Line 300-304: This study demonstrated that complete resection of necrotic bone and intraopera-tive irrigation of the maxillary sinus may provide good treatment outcomes for maxil-lary sinusitis associated with MRONJ, although not statistically significant owing to the small number of patients. In future research, a multicenter study using different treatment methods should be conducted for better generalizability of information.
Reviewer 3 Report
This is a retrospective study on sinusitis treatment associated with MRONJ. As mentioned by the same authors, there have already been published articles on this topic with a greater sample size. The surgical protocol is undetailed. The authors concluded that "This study demonstrated that irrigation therapy was effective in treating maxillary sinusitis associated with MRONJ" but the statistical analysis did not confirm this affirmation significantly.
Author Response
To Reviewer #3
This is a retrospective study on sinusitis treatment associated with MRONJ. As mentioned by the same authors, there have already been published articles on this topic with a greater sample size. The surgical protocol is undetailed. The authors concluded that "This study demonstrated that irrigation therapy was effective in treating maxillary sinusitis associated with MRONJ" but the statistical analysis did not confirm this affirmation significantly.
(Reply)
As the Reviewer pointed out, this study has a small number of cases and there is no statistical evidence that intraoperative maxillary sinus irrigation was effective in the healing process of maxillary sinusitis associated maxillary MRONJ. We would like to increase the number of cases and consider it in the future. The description about MRONJ surgery method was added.
Line 133-136: All patients underwent removal of necrotic bone and the surrounding healthy bone, and the wound was closed with a mucoperiosteal flap. If necessary, the perios-teum on the buccal side was incised and then primarily sutured. No cases were using buccal fat pad pedicle flap.
Reviewer 4 Report
Dears authors,
congratulations to this important data in treatment of maxillary MRONJ.
The reference list of the manuscript contains 43 titles, and is without inappropriate self-citations. One reference are elder than 20 years. The manuscript is clear, with a good rate of novelty and significance. The manuscript present scientific resound and the design appropriate to test the hypothesis. The methods and software are clear described, with sufficient details to permit another researcher to reproduce the results. All aspects regarding the figures/tables/images are appropriate, and they are easy to interpret and understand. The presentation and the analyzed date are written in proper way. The presentation of the results are at high standard, with appropriate statistics. The results offer a development in the present knowledge, are significant, and are suitable interpreted.
LL25: Complete healing of MRONJ was obtained in 29 of 34 patients (85.3%).
What is the criteria for complete healing? Does it correspond to the defined category (i) LL 86? If not, please clarify the term.
LL146: We examined the differences in the outcomes of maxillary sinusitis between 21 patients who healed or improved and 12 patients who remained unchanged or worsened.
Regarding the given patient characteristics in table 1, it seems, that one patient is missing, or?
LL266: For sure intraoperative irrigation contributes decreasing the mass of inflammatory tissue, mediators, and bone fragments. From my point of view and experience, healing of sinusitis in association with MRONJ is only achievable, if the Ostium to the nasal cavity is passable. Therefore, it remains for me still unclear, whether a single irrigation procedure will establish a level of low inflammatory tissue. Have you prescribed any other treatment? Nasal spray (e.g. saline, xylometazoline?)
Did you have any secondary oro-antral-fistulas after removal of the necrotic bone covering the sinus? What is your standard procedure concerning rehabilitation with prosthesis?
Author Response
To Reviewer #4
- LL25: Complete healing of MRONJ was obtained in 29 of 34 patients (85.3%). What is the criteria for complete healing? Does it correspond to the defined category (i) LL 86? If not, please clarify the term.
(Reply)
Complete healing means that all symptoms, including bone exposure, were resolved, as described in Materials and Methods (Line 98-99).
- LL146: We examined the differences in the outcomes of maxillary sinusitis between 21 patients who healed or improved and 12 patients who remained unchanged or worsened. Regarding the given patient characteristics in table 1, it seems, that one patient is missing, or?
(Reply)
Line 170: I'm sorry that the numbers in the text are incorrect. “12” was corrected to “13”.
- LL266: For sure intraoperative irrigation contributes decreasing the mass of inflammatory tissue, mediators, and bone fragments. From my point of view and experience, healing of sinusitis in association with MRONJ is only achievable, if the Ostium to the nasal cavity is passable. Therefore, it remains for me still unclear, whether a single irrigation procedure will establish a level of low inflammatory tissue. Have you prescribed any other treatment? Nasal spray (e.g. saline, xylometazoline?)
(Reply)
The sentence “However, since there were cases in which maxillary sinusitis could not be cured with just one intraoperative irrigation, so it was considered necessary to continue some kind of treatment for maxillary sinusitis in the future.” was added (Line 281-284).
- Did you have any secondary oro-antral-fistulas after removal of the necrotic bone covering the sinus? What is your standard procedure concerning rehabilitation with prosthesis?
(Reply)
There were no cases of residual oro-antral fistula (Line 149).
Round 2
Reviewer 2 Report
The authors followed the reviewers' suggestions, the manuscript appears improved in several parts and is assessable for publication.
Reviewer 3 Report
None
This manuscript is a resubmission of an earlier submission. The following is a list of the peer review reports and author responses from that submission.